High antibody titres induced by protein subunit vaccines using Mycobacterium ulcerans antigens Hsp18 and MUL_3720 with a TLR-2 agonist fail to protect against Buruli ulcer in mice

Mangas Kirstie M. 1
Tobias Nicholas J. 2 3
Marion Estelle 4 4 5
Babonneau Jérémie 4 5
Marsollier Laurent 4 5
Porter Jessica L. 1
Pidot Sacha J. 1
Wong Chinn Yi 1
Jackson David C. 1
Chua Brendon Y. bychua@unimelb.edu.au 1
Stinear Timothy P. tstinear@unimelb.edu.au 1
1 Department of Microbiology and Immunology, Peter Doherty Institute for Infection and Immunity, University of Melbourne , Melbourne , Victoria , Australia
2 Johann Wolfgang Goethe Universität Frankfurt am Main , Frankfurt , Germany
3 LOEWE Centre for Translational Biodiversity in Genomics (TBG) , Frankfurt , Germany
4 Université de Nantes , Nantes , France
5 Université d’Angers , Angers , France
Flores-Valdez Mario Alberto
Electronic publication date: 2020 Aug 7
Publication date: 2020
Volume: 8
Electronic Location ID: e9659
Received 2020 Feb 19; Accepted 2020 Jul 14
Copyright: ©2020 Mangas et al.
Copyright year: 2020
Copyright holder: Mangas et al.
License: This is an open access article distributed under the terms of the Creative Commons Attribution License, which permits unrestricted use, distribution, reproduction and adaptation in any medium and for any purpose provided that it is properly attributed. For attribution, the original author(s), title, publication source (PeerJ) and either DOI or URL of the article must be cited.
License URL: https://creativecommons.org/licenses/by/4.0/

Keywords: Mycobacterium ulcerans, Buruli ulcer, Vaccination, Mycobacterium, Subunit vaccine, ELISA, Antibody

Funding: National Health and Medical Research Council of Australia GNT1105525 This work was supported by the National Health and Medical Research Council of Australia (GNT1105525). The funders had no role in study design, data collection and analysis, decision to publish, or preparation of the manuscript.

==============================
Background

Mycobacterium ulcerans is the causative agent of a debilitating skin and soft tissue infection known as Buruli ulcer (BU). There is no vaccine against BU. The purpose of this study was to investigate the vaccine potential of two previously described immunogenic M. ulcerans proteins, MUL_3720 and Hsp18, using a mouse tail infection model of BU.

Methods

Recombinant versions of the two proteins were each electrostatically coupled with a previously described lipopeptide adjuvant. Seven C57BL/6 and seven BALB/c mice were vaccinated and boosted with each of the formulations. Vaccinated mice were then challenged with M. ulcerans via subcutaneous tail inoculation. Vaccine performance was assessed by time-to-ulceration compared to unvaccinated mice.

Results

The MUL_3720 and Hsp18 vaccines induced high titres of antigen-specific antibodies that were predominately subtype IgG1. However, all mice developed ulcers by day-40 post-M. ulcerans challenge. No significant difference was observed in the time-to-onset of ulceration between the experimental vaccine groups and unvaccinated animals.

Conclusions

These data align with previous vaccine experiments using Hsp18 and MUL_3720 that indicated these proteins may not be appropriate vaccine antigens. This work highlights the need to explore alternative vaccine targets and different approaches to understand the role antibodies might play in controlling BU.

Introduction

Buruli ulcer (BU) is a disease caused by Mycobacterium ulcerans. M. ulcerans infects subcutaneous tissue and commonly presents as a skin nodule (in Africa) or papule (in Australia), sometimes accompanied by redness; however, oedema is another common initial presentation. As the disease progresses the skin around the infected area breaks down and an ulcer develops (Guarner et al., 2003; Vincent et al., 2014). Ulcers typically present with deep undermined edges and have a necrotic core comprised of slough of bacteria, dead skin and immune cells (Hayman & McQueen, 1985; Oliveira et al., 2005). Infections are rarely fatal but untreated ulcers can destroy fat tissue, blood vessels, muscles and bone (Woodring et al., 1986; Van der Werf et al., 1999).

BU endemic areas are focused in certain rural regions across west, sub-Saharan and central Africa, including Nigeria, Ghana, Togo, Cameroon, Benin, Democratic Republic of Congo and Côte d’Ivoire. The disease also occurs in the South-East of Australia (Loftus et al., 2018; Organization, 2020; Simpson et al., 2019; Bratschi et al., 2013). The disease can affect all age groups and ethnicities (Omansen et al., 2019a).

M. ulcerans is a slow-growing bacterium, with a doubling time of greater than 48 h. As such, symptoms of BU can take months to appear after primary infection. If diagnosed early, BU can be treated effectively by combination antibiotic therapy (Sarfo et al., 2010). Unfortunately, in many cases the disease can initially be misdiagnosed as other more common skin infections (Van der Werf et al., 2005; Guarner, 2018). Delayed diagnosis and treatment can lead to extensive lesions that leave victims with life-long disfigurement and disability. Reparative surgery is often required for severe cases (Herbinger et al., 2008). A retrospective study in Australia showed that most diagnoses (87%) occurred once ulceration has been reached (Boyd et al., 2012) and in Ghana 66% cases were diagnosed with active lesions (Amofah et al., 2002). There is currently no vaccine for BU and no distinct mechanism of transmission. Furthermore, treatment can be difficult to access for those in rural areas. Thus, there is a need to develop an effective vaccine to protect those particularly in highly endemic areas.

The M. bovis ‘BCG’ vaccine has been shown to delay the onset of BU symptoms and decrease bacterial load in both experimental animal BU infection models and in studies of human populations (Tanghe et al., 2001; Tanghe et al., 2007; Phillips et al., 2015; Group, 1969; Smith et al., 1976). Therefore, the BCG vaccine is the benchmark for assessing potential M. ulcerans vaccines. Some studies have assessed the efficacy of putative BU vaccines although none have reached clinical trials (Tanghe et al., 2001; Tanghe et al., 2007; Tanghe et al., 2008; Coutanceau et al., 2006; Roupie et al., 2014; Bolz et al., 2015; Bolz et al., 2016; Fraga et al., 2012; Watanabe et al., 2015; Hart, Hale & Lee, 2016; Converse et al., 2011; Hart, Hale & Lee, 2015; Hart & Lee, 2016; Trigo et al., 2013). All these vaccines were tested in murine challenge models and were not capable of preventing the eventual onset of disease.

One approach to vaccination is to use antigens specific for a particular pathogen, e.g., certain proteins(s) that are recognized by the immune system and induce neutralizing antibodies (Siegrist, 2018; Plotkin, 1999). Vaccines that utilize only the immune stimulant, such as a protein rather than the whole pathogen, are less likely to induce adverse reactions, therefore could be administered to immunocompromised individuals (Siegrist, 2018; Clem, 2011). For rapid immune recognition these protein antigens would ideally be cell surface associated. Two M. ulcerans proteins MUL_3720 and Hsp18 have been identified as potential candidates for vaccine antigens. Hsp18 is a cell–surface associated protein, with a role in biofilm formation (Pidot et al., 2010a). M. ulcerans-infected individuals produce antibodies against Hsp18 (Pidot et al., 2010b; Diaz et al., 2006). MUL_3720 is a highly expressed cell-wall associated protein with a putative role in cell-wall biosynthesis (Vettiger et al., 2014) and MUL_3720-specific antibodies can be identified in M. ulcerans infected individuals (Dreyer et al., 2015).

Protein antigens generally require an adjuvant to boost immunogenicity and shape immune responses (Awate, Babiuk & Mutwiri, 2013; Coffman, Sher & Seder, 2010). A known Toll-like Receptor (TLR)-2 ligand, R4Pam2Cys, has been found to increase antigen uptake, increase dendritic cell trafficking to lymph nodes and enhance antibody production against antigens derived from pathogens including influenza and hepatitis C in murine models (Chua et al., 2014; Chua et al., 2011; Chua et al., 2012; Christiansen et al., 2018). Given BU is a disease where the bacteria can be both extracellular and intracellular (Torrado et al., 2007), the ability of R4Pam2Cys to robustly engage multiple arms of the adaptive immune system may be beneficial for a BU subunit vaccine. In a recent murine vaccination study, R4Pam2Cys in conjunction with the enoyl reductase domain from the mycolactone biosynthesis machinery, improved protection against Buruli ulcer compared to no vaccine (Mangas et al., 2020).

The proteins, Hsp18 and MUL_3720 have also been explored in other vaccine studies (Bolz et al., 2015; Bolz et al., 2016). These studies incorporate alternative adjuvants, such as virus replicon particles and EM048, Alum and Sigma adjuvant system in a murine footpad challenge model (Bolz et al., 2015; Bolz et al., 2016). This study focussed on inducing protection through TLR-2 signaling in a murine tail infection model. At the time this study was initiated, no other research using MUL_3720 and Hsp18 in a vaccine had been published. This study was the foundation for the development of the recent low-dose murine tail infection model of BU (Mangas et al., 2020).

The aim of this study was to try to develop a preventative vaccine against BU, comprising two highly expressed cell-wall associated proteins, MUL_3720 or Hsp18, bound to an R4Pam2Cys-based lipopeptide adjuvant evaluated via a murine tail infection model of BU.

Materials & Methods

Strains and culture conditions

Escherichia coli Rosetta2 containing plasmid pET30b-Hsp18 (strain TPS681) or pDest17-MUL_3720 (strain TPS682) was grown at 37 °C in Luria-Bertani (LB) broth (Difco, Becton Dickinson, MD, USA) supplemented with 100 µg/ml ampicillin (Sigma-Aldrich, USA) or 50 µg/ml kanamycin to express 6xHIS-tagged Hsp18 or MUL_3720 recombinant protein. Mycobacterium ulcerans (strain Mu_1G897 from French Guiana (De Gentile et al., 1992)) was grown at 30 °C in 7H9 broth or 7H10 agar (Middlebrook, Becton Dickinson, MD, USA) supplemented with oleic acid, albumin, dextrose and catalase growth supplement (OADC) (Middlebrook, Becton Dickinson, MD, USA), and 0.5% glycerol (v/v). M. bovis BCG (strain Sanofi Pasteur) used for vaccinations was grown at 37 °C in 7H9 broth or 7H10 agar supplemented with OADC. Mycobacterial colony counts from cultures or tissue specimens were performed using spot plating as previously described (Wallace et al., 2017).

Recombinant protein expression

Overnight cultures of strains TPS681and TPS682 were diluted to OD600 = 0.05 in LB broth. Each culture was incubated at 37 °C with shaking at 200 rpm until OD600 = 0.6–0.7, then 1 mM IPTG (Isopropyl b-D-1-thiogalactopyr-anoside) was added to induce protein expression. The cells were incubated for a further four hours to express the protein. To harvest the protein, cells were resuspended in wash buffer (8 M urea, 150 mM sodium chloride, 10% glycerol) and sonicated at amplitude 60 (QSonica Ultrasonic Liquid Processor S-4000, Misonix) until the solution turned clear. The lysate was filtered with a 0.22 µM filter (Millipore) to remove cellular debris and the protein was column-purified using anti-histidine resin (ClonTech). The resin was washed ten times with 10x column volumes of wash buffer mixed with an increasing proportion of tris buffer (20 mM Tris–HCl, 150 mM sodium chloride, 10% glycerol) until the column was washed with only tris buffer. The resin was washed a further two times with tris buffer containing 20 mM imidazole. Protein was eluted in tris buffer containing 200 mM imidazole and dialysed in phosphate buffered saline (PBS) before concentration using a 3K MWCO PES concentration column (Pierce). Protein was endotoxin purified using Triton X-114 phase separation until less than 0.1 endotoxin unit/ml (detectable limit), measured by PierceTM LAL choromogenic endotoxin quantitation kit (ThermoFisher) as per manufacturer’s instructions.

Sodium dodecyl sulphate polyacrylamide gel electrophoresis (SDS-PAGE)

Samples were denatured in an equal volume of 2 x sample loading buffer (40% (v/v) 0.5M Tris-HCL pH 6.8, 10% glycerol, 1.7% (w/v) SDS, 10% 2-β-mercaptoethanol, 0.13% (w/v) bromophenol blue in distilled water) at 100 °C for 5 min. Ten microlitres of each sample and SeeBlue® Plus2 pre-stained protein standard (Invitrogen) were loaded into a 0.5 mm 12% polyacrylamide gel under reducing conditions, as previously described (Laemmli, 1970). The gel was run in running buffer (0.3% (w/v) Tris, 1.44% (w/v) glycine and 0.1% (w/v) SDS in distilled water) for 1 h at 150 volts (Mini-protean vertical electrophoresis cell, Bio-Rad). The gels were stained in Coomassie stain (45% methanol, 10% acetic acid 0.25% (w/v) Coomassie brilliant blue in distilled water) for 1 h and destained in Coomassie destain (33% Methanol, 10% acetic acid, 60% distilled water) until the protein bands could be identified.

Western Blotting

Proteins were separated on a 12% polyacrylamide gel as per the method for SDS-PAGE. After separation proteins were transferred to a nitrocellulose membrane in tris-glycine transfer buffer (1.5 mM Tris, 12mM glycine, 15% methanol (v/v) in distilled water) for 1 h at 100 volts (Mini Trans-Blot Cell, Bio-Rad). The nitrocellulose membrane was blocked in blocking buffer (5% (w/v) skim milk powder and 0.1% Tween-20 in PBS) overnight at 4 °C. The membrane was incubated in blocking buffer containing anti-6xHIS-HRP antibody (Roche Applied Science) at 1:500 dilution. The membrane was washed in PBS containing 0.1% Tween-20 and then exposed to developing solution (Western Lighting Chemiluminescence kit, Perkin Elmer) according to manufacturer’s guidelines. Chemiluminescence was detected using an MF ChemiBIS gel imaging system (DNR Bio-Imaging Systems).

Analysis of electrostatic interaction between protein antigen and lipopeptide formulations

The association between each protein and R4Pam2Cys was measured by mixing 25 µg of protein with increasing amounts of lipopeptide in 50 µl PBS in a 96-well plate (Nunc, Thermo Scientific). The formation of protein-lipopeptide complexes through electrostatic interaction was measured by an increase in light absorbance. Plates were read at dual wavelengths of 505 and 595 nm on plate reader (LabSystems Multiskan Multisoft microplate reader).

Lipopeptide vaccine preparation

Each vaccine dose contained 25 µg protein added to R4Pam2Cys at a ratio of 1:5 mole of protein to lipopeptide. PBS was added to a final volume of 100 µl and the combination sonicated in a water bath for 30 s. Control vaccine preparations were made containing 25 µg protein alone or R4Pam2Cys lipopeptide alone and sonicated before administration.

Ethics statement for animal experiments

All animal experiments were performed in full compliance with national guidelines (articles R214-87 to R214-90 from French “rural code”) and European guidelines (directive 2010/63/EU of the European Parliament and of the council of September 22, 2010 on the protection of animals used for scientific purposes). All protocols were approved by the Ethics Committee of region Pays de la Loire under protocol nos. CEEA 2009.14 and CEEA 2012.145. Animals were maintained under specific pathogen-free conditions in the animal house facility of the Centre Hospitalier Universitaire, Angers, France (agreement A 49 007 002). Six-week old female C57BL/6 and BALB/c mice were obtained from Charles River Laboratories (Saint-Germain-Nuelles, France) and housed at CHU Angers. Food and water were given ad libitum. Animals were euthanized by inhalation of CO2 gas, delivered using a gradual fill method in a chamber containing the animal. All personnel using this technique were appropriately trained to operate the equipment, evaluate animal vital signs and confirm death.

Vaccination of animals

The synthesis and purification of the branched cationic lipopeptide, R4Pam2Cys, was performed as previously described (Chua et al., 2011; Sekiya et al. 2017; Wijayadikusumah et al., 2017). Each vaccine dose contained 25 µg protein formulated in PBS with R4Pam2Cys at a 1:5 molar ratio of protein to lipopeptide in a final volume of 100 µl. The protein alone control formulation contained 25 µg protein per dose diluted in PBS. The R4Pam2Cys alone formulations contained the same amount of lipopeptide used in each of the protein + adjuvant formulations, calculated by the 1:5 molecular ratio (with the omission of the protein from the solution). The R4Pam2Cys alone formulations were diluted to the correct concentration in PBS. Live-attenuated M. bovis BCG strain ‘Sanofi Pasteur’ was grown to log phase and stored at −80 °C in 20% glycerol until use. Bacteria were washed with PBS and resuspended in 200ul, before administration at 4.7 × 105 bacteria per dose. All vaccines and control formulations were sonicated for 5 min in a waterbath sonicator before being administered.

For vaccination using R4Pam2Cys, animals were inoculated subcutaneously at the base of tail (100 µl per dose at 50 µl per flank) and boosted 21 days later with the same formulations. Mice vaccinated with approximately 1 × 103 CFU M. bovis BCG resuspended in PBS at the base of tail (100 µl per dose at 50 µl per flank).

M. ulcerans challenge

Mice were challenged on day 35 by subcutaneous injection on the tail with 1 × 104 CFU M. ulcerans (Mu_1G897) resuspended in 50 µl PBS. Mice were allowed to recover and monitored for up to 40 days after infection and euthanised when tail ulceration was observed wherein sera were obtained for immunological analysis.

Serum antibody titre measurements

Serum was prepared from blood obtained from mice at day 0, day 18, day 33 and day 63. Antibody titres were measured using enzyme linked immunosorbent assay (ELISA) as per methods described in Chua et al. (2011). Briefly, ELISA plates (Nunc, Thermo Scientific) were coated overnight with 5 µg/ml protein diluted in PBSN3 and blocked with BSA10PBS for 2 h at room temperature. Plates were washed with PBS containing 0.05% Tween-20 (PBST). Neat sera were sequentially diluted in BSA5PBST and incubated at room temperature for 6 h. Bound antibody was detected by adding horse radish peroxidase conjugated rabbit anti-mouse IgG (Dako, Glostrup, Denmark) or rat anti-mouse IgM, IgG1, IgG2a, IgG2b or IgG3 antibodies (Southern Biotech, USA) at a concentration of 1:400 in BSA5PBST for 2 h. Plates were developed with developing solution (hydrogen peroxide, citric acid and ABTS) and incubated for 10-15 min with gentle agitation to observe a colour change. The reaction was stopped with 50 mM sodium fluoride. Plates were read at dual wavelengths of 505 and 595 nm on plate reader (LabSystems Multiskan Multisoft microplate reader). The titers of antibody are expressed as the reciprocal of the highest dilution of serum required to achieve an OD600 of 0.2.

Statistical analysis

Graphpad Prism software (GraphPad Software v7, CA, USA) was used to perform statistical analyses on the antibody titre. Antibody titres were analysed using two-way ANOVA with Tukey’s correction for multiple comparisons. The time-to-ulceration data were displayed as a Kaplan–Meier plot and statistical significance was determined using a Log-Rank (Mantel-Cox) test. For all tests *p <0.05, **p <0.01 and ***p <0.001 and **** p <0.0001 were considered statistically significant.

Results

MUL_3720 and Hsp18 have previously been shown to be cell-wall associated and immunogenic in humans (Pidot et al., 2010b; Diaz et al., 2006; Vettiger et al., 2014; Dreyer et al., 2015). The adjuvant Pam2Cys has been shown to induce strong antibody responses to proteins from infectious agents such as influenza and hepatitis C in mice (Wijayadikusumah, 2017; Tan et al., 2012; Mifsud et al., 2016). Therefore, this study measures the ability of MUL_3720 and Hsp18 based vaccines, incorporating the adjuvant Pam2Cys, to generate protein-specific antibodies and to protect against BU.

Recombinant MUL_3720 and Hsp18 both bound to R4Pam2Cys

Recombinant MUL_3720 and Hsp18, expressed from inducible E. coli expression vectors, were prepared for use as antigens in the vaccine formulations (Table S1). Purification of the recombinant proteins was confirmed by SDS-PAGE and Western blot analyses of the eluate (Fig. 1). DLS analysis was then performed to identify whether recombinant MUL_3720 or Hsp18 would electrostatically bind to either the positively charged lipopeptide adjuvant R4Pam2Cys, or its negatively charged counterpart, E8Pam2Cys. The optical density of solutions containing these constituents at a wavelength of 450 nm (OD450) is related to the particle size of molecules in solution, reflecting the strength of the ionic interaction between protein and lipopeptide (Chua et al., 2011). MUL_3720 preferentially bound to R4Pam2Cys compared to E8Pam2Cys (Fig. 2A, Table S2). This is shown as a gradual increase in optical density following the addition of increasing amounts of R4Pam2Cys to a constant amount of MUL_3720. At a 5-fold molar excess of protein to lipopeptide the OD 450plateaued, suggesting MUL_3720 bound most strongly to R4Pam2Cys at a 1:5 protein to lipopeptide ratio. Conversely, when E8Pam2Cys was added to MUL_3720 the optical density remained static and did not increase with increasing lipopeptide concentrations, indicating a lack of binding. Hsp18 also appeared to bind preferentially to R4Pam2Cys and also at a 1:5 ratio of Hsp18 to R4Pam2Cys (Fig. 2B). Therefore, two protein-adjuvant formulations were prepared using MUL_3720 with R4Pam2Cys and Hsp18 with R4Pam2Cys, both at a 1:5 protein to lipopeptide molar ratio.

Figure 1 SDS-PAGE and Western Blot Analysis of purified recombinant MUL_3720 and Hsp18 proteins.

(A) SDS-PAGE of MUL_3720 protein elution (containing 10 µg protein) shows a band 36 kDa. (B) SDS-PAGE of Hsp18 protein elution (containing 10 µg protein) shows a band ∼18 kDa. (C) Protein in the final MUL_3720 elute was analysed by Western blot using an anti-6xHIS-tag antibody to detect the presence of a single band corresponding to the band as the SDS- PAGE analysis. (D) Protein in the final Hsp18 elute was analysed by Western Blot using an anti-6xHIS-tag antibody to detect the presence of a single band corresponding to the 18 kDa band as the SDS-PAGE analysis.

Figure 2 Recombinant MUL_3720 and Hsp18 protein formulation linked with R_4Pam_2Cys.

To analyse the formation of antigen-lipopeptide complexes, a constant amount of antigen (A) MUL_3720 (25 µg) and (B) Hsp18 (25 µg) was mixed with lipopeptide at different protein:lipopeptide molar ratios in 50 µl of PBS. These graphs depict the absorbance values of these solutions at an optical density of 450 nm (OD_450). In these assays either R_4Pam_2Cys or E_8Pam_2Cys lipopeptides were added to the proteins at increasing amounts. The addition of R_4Pam_2Cys is depicted with black circles and the addition of E_8Pam_2Cys is depicted with grey squares. An increase in absorbance in correlation to an increase in lipopeptide was indicative of protein binding to lipopeptide.

Vaccination induced strong protein-specific antibody responses

Prior to challenge with M. ulcerans, the ability of the vaccine candidates to generate murine immune responses was assessed. ELISAs were utilized to measure the antibody (IgG) titres in sera obtained from two strains of mice (BALB/c and C57BL/6) immunized with either MUL_3720 + R4Pam2Cys or Hsp18 + R4Pam2Cys after the primary vaccination dose (day 18) and a secondary dose (day 33) (Fig. 3, Table S3).

Figure 3 Antibody titres from BALB/c and C57BL/6 mice immunized with recombinant MUL_3720 or Hsp18 linked to R4Pam2Cys lipopeptide adjuvant.

MUL_3720-specific antibody titres from (A) BALB/c and (B) C57BL/6 mice. Mice were vaccinated with protein alone (MUL_3720) (grey circles), recombinant protein + R4Pam2Cys (blue circles), R4Pam2Cys alone (clear circles) and M. bovis BCG (black circles). The error bars represent standard deviation (SD) (n = 7). A separate ELISA was performed to measure Hsp18-specific antibody titres in (C) BALB/c and (D) C57BL/6 mice. Mice were vaccinated with protein alone (Hsp18) (grey circles), recombinant protein + R4Pam2Cys (blue circles), R4Pam2Cys alone (clear circles) and M. bovis BCG (black circles). The error bars represent SD (n = 7). Responses for day-63 post M. ulcerans challenge highlighted by grey shading. IgG isotypes (IgG1, IgG2a, IgG2b and IgG3) were quantified from BALB/c mice immunized with (E) MUL_3720 + R4Pam2Cys and (F) Hsp18 + R4Pam2Cys. Mice were vaccinated with protein antigen alone (either MUL_3720 or Hsp18) (clear circles), protein + R4Pam2Cys (grey circles) and BCG (black circles). Results are shown as zero if below detectable limits. The error bars represent SD (n = 4). The null hypothesis (no difference in mean antibody responses between treatment groups) was rejected at *p < 0.05, **p < 0.01, ***p < 0.001 or ****p < 0.0001.

Vaccination with MUL_3720 recombinant protein alone or MUL_3720 + R4Pam2Cys were capable of inducing MUL_3720-specific antibody titres in both BALB/c and C57BL/6 strains of mice (Figs. 3A, 3B). Primary vaccination with MUL_3720 protein alone induced MUL_3720-specific antibody responses that significantly increased following a vaccine boost (p < 0.0001 and p = 0.0005 for BALB/c and C57BL/6, respectively). Additionally, MUL_3720 + R4Pam2Cys generated MUL_3720 specific antibody responses after primary vaccination (p  <  0.0001 in BALB/c and C57BL/6), which were increased after the secondary boost (p <0.0001 in BALB/c and p = 0.0035 in C57BL/6). The titres after the boost in particular were greater than MUL_3720 alone vaccination (p  < 0.0001 in BALB/c and p = 0.0075 in C57BL/6). Mice that were not vaccinated with recombinant MUL_3720 (R4Pam2Cys alone and BCG) did not have an increase in MUL_3720-specific antibodies compared to naïve mice.

Vaccination with Hsp18 recombinant protein alone or Hsp18 + R4Pam2Cys induced Hsp18-specific antibody titres in both strains of mice (Figs. 3C, 3D). Vaccine boost with Hsp18 recombinant protein alone induced significantly higher Hsp18-specific antibody responses in BALB/c mice compared to a single vaccination with Hsp18 protein (p  <  0.0001). Boosting with protein alone in C57BL/6 did not significantly increase antibody titres. Hsp18 + R4Pam2Cys induced Hsp18-specific antibody responses in both mouse strains after primary vaccination (p  <  0.0001 in BALB/c and C57BL/6) and the Hsp18-specific antibody titre significantly increased after booster vaccination (p  <  0.0001 in BALB/c and p = 0.0006 in C57BL/6). In all strains, the antibody titres induced by Hsp18 + R4Pam2Cys were significantly higher than vaccination with Hsp18 protein alone (p  <  0.0001 in BALB/c and C57BL/6) (Figs. 3C, 3D) with negligible levels of antibodies seen in mice vaccinated with only R4Pam2Cys, or BCG.

Measurement of IgG antibody subtypes following MUL_3720 + R4Pam2Cys and Hsp18 + R4Pam2Cys vaccination

Quantifying levels of IgG antibody shows that the predominant isotypes produced by MUL_3720 were IgG1 and IgG2b (Fig. 3E) with no significant difference between these isotype titres. The antibody titres for both isotypes were highest prior to infection with M. ulcerans (day 33) and decreased after infection by day 63. This vaccine was capable of inducing IgG2a antibodies, which was detected also on day 33, however in smaller amounts than IgG1 (p = 0.0300; Fig. 3E).

Similar to vaccination with MUL_3720, Hsp18 was also capable of inducing strong IgG antibody titres. The predominant isotype was IgG1 which Hsp18 + R4Pam2Cys elicited more than any other isotype including IgG2a (p = 0.0317) and IgG2b (although, not significant) (Fig. 3F). Again, these titres were highest at day 33 and decreased significantly after infection on day 63. This trend was also observed after vaccination with Hsp18 alone (p < 0.0001 vs IgG2a and p = 0.0001 vs IgG2b, respectively at day 33).

MUL_3720 + R4Pam2Cys and Hsp18 + R4Pam2Cys do not protect against the onset of BU

As both vaccines were capable of inducing protein-specific antibody responses, they were tested in a murine challenge model to measure their protective efficacy. Efficacy was measured by time delay to the onset of ulceration in a mouse tail infection model. There is a progression of clinical symptoms for Buruli ulcer in this model (Fig. 4). Once ulceration has been reached the disease would likely continue until the tail became necrotic. Therefore, the experimental endpoint was deemed to be the point of ulceration.

Figure 4 Progression of BU in the murine tail infection model over time.

(A) Healthy mouse tail. (B) Appearance of a small sign of redness at the site of tail infection. (C) Oedema surrounding the initial site of redness. (D) Tail lesion at the point of ulceration. This is typically identified by excessive oedema and redness at the site of imminent ulceration. Mice were culled before ulcerative lesions appeared.

After the scheduled vaccinations, mice were challenged via subcutaneous tail inoculation with 1 × 104 CFU of M. ulcerans and observed for up to 40 days. In BALB/c and C57BL/6 mice there was no significant difference between the time to ulceration between control mice (mice not vaccinated with recombinant protein, such as R4Pam2Cys alone and BCG) and mice vaccinated with either MUL_3720 + R4Pam2Cys or Hsp18 + R4Pam2Cys (Figs. 5A, 5B). There was also no significant difference in the time to ulceration between mice that were vaccinated with MUL_3720 + R4Pam2Cys or Hsp18 + R4Pam2Cys and BCG, the benchmark for mycobacterial vaccine efficacy. Signs of infection in mice were visible by day 63 (Tables S4 and S5) and all mice reached ulceration by day 75, 40 days post-M. ulcerans challenge (Figs. 5A, 5B).

Figure 5 Vaccine performance using murine tail infection model of BU.

Survival analysis showing the time taken (days) for each mouse to reach ulceration for different vaccination groups post M. ulcerans challenge. (A) BALB/c mice (n = 7) and (B) C57BL/6 mice (n = 7). The null hypothesis (no difference in mean antibody responses between treatment groups) was rejected if ∗p < 0.05, ∗∗p < 0.01, ∗∗∗p < 0.001 or ∗∗∗∗p < 0.0001.

Antibody titres do not correlate with protection against M. ulcerans

High antibody titres were observed in all mice vaccinated with either recombinant MUL_3720 or Hsp18, particularly in the secondary response after booster vaccination (Figs. 3A–3D) prior to M. ulcerans challenge. However, mice vaccinated with protein alone or protein plus lipopeptide adjuvant all succumbed to infection by day 75. The sera from mice at the day 63 was used to quantify antibody titres during infection. At day 63 all mice still had detectable protein-specific antibodies against the recombinant protein with which they were vaccinated (Figs. 3A–3D). In BALB/c mice (Figs. 3A, 3C) the antibody titres at day 63 were lower than after the secondary response prior to challenge (p < 0.0001 Hsp18 + R4Pam2Cys and not significant for MUL_3720 + R4Pam2Cys) but remained significantly higher than at day 0 (p < 0.0001 for both Hsp18 + R4Pam2Cys and MUL_3720 + R4Pam2Cys). In C57BL/6 mice (Figs. 3B and 3D), antibody titres against MUL_3720 or Hsp18 from mice vaccinated with either protein alone or protein plus lipopeptide adjuvant were also significantly decreased at day 63 compared to the secondary response at day 35 (p < 0.0001 for MUL_3720 + R4Pam2Cys and Hsp18 + R4Pam2Cys, respectively). Similar to BALB/c mice, the day 63 respective protein-specific antibodies for MUL_3720 + R4Pam2Cys and Hsp18 + R4Pam2Cys were significantly higher than at day 0 (p < 0.0001 for MUL_3720 + R4Pam2Cys and Hsp18 + R4Pam2Cys).

Challenge with M. ulcerans did not induce protein-specific antibody levels comparable to vaccination with MUL_3720 or Hsp18

MUL_3720 and Hsp18 recombinant proteins are immunogenic and capable of inducing protein-specific antibody responses after vaccination. However, only minor detectable antibody responses against either recombinant MUL_3720 or Hsp18 at day 63 (Figs. 3A–3D) were found in mice vaccinated with R4Pam2Cys alone or BCG then challenged with M. ulcerans. These responses are much lower than the protein-specific antibody responses generated from MUL_3720 or Hsp18 vaccinated mice, particularly in C57/BL6 mice (p < 0.0001) (Figs. 3A–3D). Animals from both mouse strains that were vaccinated with R4Pam2Cys alone or BCG showed no increase in protein-specific antibody responses against either recombinant MUL_3720 and Hsp18 on day 63 post-M. ulcerans challenge (Figs. 3A–3D), even though these two proteins are both expressed in M. ulcerans.

Discussion

This study aimed to develop a vaccine against M. ulcerans utilizing two previously described cell-wall associated proteins, Hsp18 and MUL_3720 (Pidot et al., 2010a; Pidot et al., 2010b; Vettiger et al., 2014; Dreyer et al., 2015). Both the MUL_3720 and Hsp18-based vaccines were capable of inducing high antibody titres, but these responses were not associated with protection (Fig. 5). Since our study was conducted, Bolz et al., also reported experimental BU vaccines incorporating Hsp18 and MUL_3720 (Bolz et al., 2015; Bolz et al., 2016). These vaccine formulations included the following adjuvants: virus replicon particles (Bolz et al., 2015), TLR-4 agonist EM048, Alum and Sigma adjuvant system (Bolz et al., 2016). Similar to the findings by Bolz et al., in our study the antibodies produced by MUL_3720 and Hsp18 in conjunction with TLR-2 agonist, R4Pam2Cys, were not able to protect against M. ulcerans challenge. This may indicate that these proteins, while strongly immunogenic, play no major role in pathogenesis, so targeting them with potentially neutralizing antibodies induced by the vaccine has no impact on disease.

Alternatively, antibodies raised by these vaccines may not have had the functional potential to control infection. In addition to antigen binding, antibodies engage via their Fc domains with Fcγ receptors (FcγR) present on innate immune cells (NK cells, monocytes, macrophages and neutrophils) to rapidly recruit the anti-microbial activity of the innate immune system. Antibodies with these functions can promote control of a pathogen through the activation of multiple effector cell functions, including Ab dependent cellular cytotoxicity, cellular phagocytosis and/or cytokine and enzyme secretion (Lu et al., 2016; Damelang et al., 2019; Chung et al., 2015). Recent research has shown that mice lacking antibodies have increased susceptibility to M. tuberculosis infection (Maglione, Xu & Chan, 2007) and non-human primates treated to deplete B cells also exhibit increased bacterial burden (Phuah et al., 2016). Despite the findings of this research, it is likely that B cells and antibody responses still play a role in controlling M. ulcerans infection in this model, albeit with different specificities. A recent study of BU-infected FVN/B mice that are capable of spontaneously healing, has identified significantly higher mycolactone-specific IgG2a antibodies in the skin compared to non-spontaneously healing mice strains BALB/c and C57BL/6 (Foulon et al., 2020). As spontaneous healing and spontaneous partial healing of BU are known uncommon occurrences in both humans (Marion et al., 2016a; O’Brien et al., 2019) and mice (Marion et al., 2016b), future research could use human BU patient cohorts and as well as mouse infection models to attempt to characterize the targets, functional and structural aspects of antibody responses that differentiate subjects able to control BU from susceptible subjects. It might then be possible to use B cell probe technologies to isolate Ag-specific memory B cells from individuals that control M. ulcerans infection and then clone the immunoglobulin gene sequences identified (McLean et al., 2017). Antigen-specific monoclonal Abs (mAbs) could then be generated and characterized for their in vitro anti-microbial activity and used in in vivo mouse passive transfer studies to determine potential use as mAb therapeutics against BU.

Another explanation for the ineffectiveness of antibodies in this study may be due to the localized immune suppression induced by the M. ulcerans toxin mycolactone at the site of infection (Fraga et al., 2011). Mycolactone diffuses into tissue surrounding the bacteria (George et al., 1999; Boulkroun et al., 2010; Baron et al., 2016). Mycolactone is a cytotoxin that modulates the function of several immune cells (Baron et al., 2016; Ogbechi et al., 2018). The toxin inhibits the Sec61 translocon, affecting T cell activation, impairing T cell responsiveness and distorting cytokine production (Boulkroun et al., 2010; Baron et al., 2016). The mycolactone-induced depletion of T cell homing to peripheral lymph nodes affects subsequent B-cell activation and migration from the lymphatics (Guenin-Mace et al., 2011). The antibodies induced by the vaccine in this study may be functional but unable to access bacteria within the infection or it may be that multiple effector cell functions have been modulated by mycolactone exposure through interference with receptor expression on key innate immune cells, rendering these cells poorly responsive to antibodies. Suppression of protein-specific antibody production in the presence of mycolactone has been observed (Shinoda, Nakamura & Watanabe, 2017). Mycolactone administered to a different location than the antigen caused no reduction to systemic antigen-specific IgG titres (Shinoda, Nakamura & Watanabe, 2017), similar to the observations from our study. Monoclonal antibodies against mycolactone have been shown to neutralize the cytotoxic activity of mycolactone in vitro indicating that mycolactone could be a viable vaccine target (Dangy et al., 2016). A recent study incorporating the enoyl reductase (ER) enzymatic domain, from the polyketide synthases that form mycolactone, has shown a correlation between ER-specific antibodies and protection against the onset of Buruli ulcer (Mangas et al., 2020). This suggests that there is a role for antibodies in BU protection, though the most effective antigenic targets may be found in the mycolactone biosynthesis pathway.

The greatest antibody responses were of the IgG1 subclass. Typical antibody responses against proteins occur via B cell isotype switching from IgM (non-specific antibody isotype) to IgG. There are 4 subclasses of IgG (IgG1, IgG2, IgG3 and IgG4) and isotype switching to predominantly IgG1 suggests refinement of immune responses to respond specifically to either MUL_3720 or Hsp18, as IgG1 is capable of binding to protein antigens (SchroederJr & Cavacini, 2010). IgG1 can also bind all forms of FcγR which is required to elicit and mediate effector immune functions as described above (Sibéril et al., 2007). The presence of IgG2 suggest further isotype switching from IgG1 to IgG2a∕b as the immune response develops. IgG2 is less effective at inducing phagocytosis and fixing complement and is more commonly associated with polysaccharide antigens. Though tests on the recombinant proteins had undetectable levels of lipopolysaccharide (LPS), there could be trace amounts from the E. coli expression vector boosting IgG2 responses. This isotype switch may not necessarily be linked to poorer outcomes, as mentioned earlier, spontaneously healing FVN/B mice produce more mycolactone-specific IgG2a than mice that do not spontaneously heal (Foulon et al., 2020). Studies analysing antibodies generated during leprosy and TB infection show a switch from IgG1 to IgG2 antibodies for leprosy and a persistence of IgG1 and IgG3 antibodies for TB (Sousa et al., 1998). As isotype switching of antibodies requires help by T helper cells, future work could therefore also incorporate studies on the effect of vaccination and subsequent M. ulcerans-infection on T cells as well as antibody responses.

In this study, all mice succumbed to infection in a relatively short period (40 days) compared to previous mouse tail infection models (Omansen et al., 2019b) and human BU, where the incubation period is estimated at 4.8 months before the onset of ulceration (Loftus et al., 2018). All BALB/c and C57BL/6 mice succumbed to infection by 40 days after MU infection, even mice that were vaccinated by M. bovis BCG. M. bovis BCG has been previously shown to delay the onset of disease on average by at least 6 weeks (Tanghe et al., 2001; Tanghe et al., 2007; Fraga et al., 2012). In this study however, there was no significant difference between mice vaccinated with either MUL_3720 or Hsp18 protein alone or with both proteins plus R4Pam2Cys. This suggests that M. bovis BCG is ineffective at protecting mice in this model of M. ulcerans vaccination. This failure to observe any protective impact of M. bovis BCG might be a reflection of the challenge strain of M. ulcerans used (strain Mu_1G897) and/or the high challenge dose used (104 bacteria). High concentrations (>104 bacteria) have not been reported in environmental sources of M. ulcerans (Fyfe et al., 2010; Stinear et al., 2000; Williamson et al., 2012; Marion, 2010; Johnson et al., 2007), consistent with the hypothesis that a relatively small bacterial inoculum is required to establish BU (Stinear et al., 2000). At the time this study was conducted the minimum infectious dose (ID50) for BU had not been determined, however the ID50 has since been identified as approximately 3 CFU (Wallace et al., 2017). Future studies could use a murine model that is more representative of this low minimum infectious dose, as we recently reported in a subsequent trial of a protein subunit vaccine using the mycolactone PKS domains (Mangas et al., 2020) (discussed below).

As mentioned earlier, two other studies have also used Hsp18 and MUL_3720 proteins in vaccine studies (29, 30). Those studies focused on the footpad challenge model whereas this study utilizes a tail infection model. The previously established BU tail challenge model (Coutanceau et al., 2006), was chosen as it decreases the impact on mouse mobility and may also prevent added trauma, inflammation or secondary infections at the challenge site (Kamala, 2007), particularly given that M. ulcerans is a slow growing pathogen and mice can endure symptoms of BU for a number of weeks (Coutanceau et al., 2006; Omansen et al., 2019b). This study was a precursor to a recently published study, that utilized a vaccine challenge model that is more representative of a natural M. ulcerans infection, reflected both in the mode of M. ulcerans entry into the subcutaneous tissue and in the dose of bacteria used for challenge (Mangas et al., 2020). Using a low-dose challenge model enabled the separation of BCG-protected mice vs unvaccinated mice, a characteristic that we did not observe in the high-dose challenge used in the current study. Thus, a low-dose murine challenge model appears more relevant for measuring BU vaccine efficacy.

Conclusions

Vaccination with either MUL_3720 or Hsp18 proteins induced high antibody titres. These responses were augmented when either protein was linked with the lipopeptide adjuvant R4Pam2Cys. However, robust antibody responses did not correlate with protection against challenge with M. ulcerans. Future work could test different M. ulcerans antigens in vaccine formulations against Buruli ulcer. As mycolactone is a key virulence factor, neutralising this toxin early in infection by targeting the PKS enzymes required for its biosynthesis could be a focus for future vaccination developments. Using a low M. ulcerans inoculum as a more realistic vaccine challenge dose is also warranted.

Supplemental Information

Supplemental Information 1 Summary of protein antigens and assay characteristics

Click here for additional data file.

Supplemental Information 2 Raw data for protein-adjuvant binding

Click here for additional data file.

Supplemental Information 3 Antibody titre raw data

Click here for additional data file.

Supplemental Information 4 Survival outcomes BALB/C mice

Click here for additional data file.

Supplemental Information 5 S urvival outcomes in vaccinated C57B6 mice

Click here for additional data file.

We thank Roy Robins-Browne for providing the pET-30b MOD plasmid used for the expression of recombinant Hsp18.

Additional Information and Declarations

Competing Interests

Author Contributions

Animal Ethics

Data Availability

Timothy Stinear is an Academic Editor for PeerJ.

Kirstie M. Mangas, Nicholas J. Tobias, Estelle Marion and Laurent Marsollier conceived and designed the experiments, performed the experiments, analyzed the data, prepared figures and/or tables, authored or reviewed drafts of the paper, and approved the final draft.

Jérémie Babonneau performed the experiments, analyzed the data, prepared figures and/or tables, authored or reviewed drafts of the paper, and approved the final draft.

Jessica L. Porter performed the experiments, authored or reviewed drafts of the paper, and approved the final draft.

Sacha J. Pidot and Timothy P. Stinear conceived and designed the experiments, analyzed the data, prepared figures and/or tables, authored or reviewed drafts of the paper, and approved the final draft.

Chinn Yi Wong performed the experiments, prepared figures and/or tables, authored or reviewed drafts of the paper, and approved the final draft.

David C. Jackson conceived and designed the experiments, analyzed the data, authored or reviewed drafts of the paper, and approved the final draft.

Brendon Y. Chua conceived and designed the experiments, performed the experiments, analyzed the data, prepared figures and/or tables, authored or reviewed drafts of the paper, and approved the final draft.

The following information was supplied relating to ethical approvals (i.e., approving body and any reference numbers):

The Ethics Committee of region Pays de la Loire provided full approval for this research under protocol nos. CEEA 2009.14 and CEEA 2012.145.

The following information was supplied regarding data availability:

The raw measurements are available as a Supplemental File.

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
