# Peer review of "High antibody titres induced by protein subunit vaccines using Mycobacterium ulcerans antigens Hsp18 and MUL_3720 with a TLR-2 agonist fail to protect against Buruli ulcer in mice"

_PeerJ, doi:10.7717/peerj.9659_

## Round 0.1 · original submission · Major Revisions

Your manuscript received some specific suggestions worth considering to provide a revised version of your work, particularly those from reviewers 1 and 2.

·

Basic reporting

no comment, except for the fact that the research group recently published work with the same lipid adjuvant R4Pam2Cys in the context of cellular immune protection in a murine animal model that was not cited in the current work (Infect Immun 2020;88(3):e00753-19) and I think it would be better to make that citation, to clarify that the adjuvant was used earlier by this group; the references made to this adjuvant are probably less relevant than their own work.

Experimental design

no comment

Validity of the findings

no comment

Additional comments

well written report; well illustrated
minor:
check line 181 µ in stead of u; line 302 signle, not plural: . . There is

Reviewer 2 ·

Basic reporting

Major issue:
The introduction and background does not give enough information on relevant topics the reader needs to know to understand the rationale of this study. Instead, the introduction dwells too much on basic BU knowledge and on the situation in Australia.
For example: Lines 52-56 is on BU transmission speculation (not relevant to understand this study) but the important information is only given in Line 71.

Not enough background on the reasoning for the choice of proteins is given.
(Line 84-86 does not have a reference where you could read up as an interested audience)
Not enough information on the reasoning for the choice of Adjuvant is given.
Information on why Antibodies could be of importance for protection against M. ulcerans infection and why this study looks at them is not given at all.

Relevant literature is partially missing. For example:
- Line 61 should additionally cite:
PLoS Negl Trop Dis. 2013 Jun 13;7(6):e2252. doi: 10.1371/journal.pntd.0002252. Print 2013. Geographic distribution, age pattern and sites of lesions in a cohort of Buruli ulcer patients from the Mapé Basin of Cameroon.
- Lines 363-370: Reference that shows that these people exist is necessary.
- Line 372-373: Reference missing
- Line 228: Reference 40 is not sufficient to show that MUL3720 is immunogenic

The Title of the manuscript could be improved, it only describes part of the study that is described (the positive aspect) and fails to mention that the vaccine is not protective.

Minor issues:
The Figures are mostly clear and well described. In Figure 3 E and F the titres for BCG could be removed to make the figure less busy. Those titres are baseline/background as can be seen in Figure 3 A-D, no need to plot them again. All information of Figure 6 is already present in Figure 3. This could be presented in a less repetitive way.

Most of the Article is easy to read and to follow. A few details should be changed:
Line 11: Affiliation is incomplete
Line 71: What do you mean with protective treatment? A vaccine?
Line 384: replace “to” with “than”
Line 229: Type on mice at the end of line

Experimental design

Major issue:
Overall, the article mostly reports data that have already been published by other authors in:
PLoS Negl Trop Dis. 2016 Feb 5;10(2):e0004431. doi: 10.1371/journal.pntd.0004431. eCollection 2016 Feb.
Vaccination with the Surface Proteins MUL_2232 and MUL_3720 of Mycobacterium ulcerans Induces Antibodies but Fails to Provide Protection against Buruli Ulcer.

Albeit with a different adjuvant and in a tail-infection model instead of the more commonly used foot pad model, but without clearly mentioning the above publication and without giving a rationale why the authors took the same two antigens again (Hsp18=MUL2232), combined with a different adjuvant and repeated mouse experiments, without explicitly addressing/discussing the similarities/differences between the two studies (not even in the Discussion).

Considering the above publication, I would question the knowledge gap filled (it is not identified in the text) with the study and the ethical standards (to some degree).

The methods are generally descried appropriately. However, there could be improvements, for example:
- The timing of the immunizations and subsequent infection is unclear. From the M&M part it seems there were 14days between the second immunization and the challenge and all mice were sacrificed by day 75 (35 days of infection max.). However, Line 315 indicates day 75 corresponds to day 30 of infection? What is it?
- Line 105: Where does that strain come from? (Reference?)
- Line 206/207: 5 ug of protein per well? Per plate?
- Line 217 following: How were the Antibody titres calculated?
- Line 395-396: If LPS-testing was done it should be mentioned in the M&M
- How did you determine IgG types? (different secondary Ab’s? If so, you should give the references)

Although the four stages of the disease progression in the tail model for BU are illustrated, it remains unclear whether there was a grading system used and at what point animals were sacrificed. This could be made more clear. Did you assess daily? And if so, where are the raw data for that?

Validity of the findings

I think the findings of the study are valid and meaningful, if not entirely novel. (See comment in Section 2).

Underlying data have been provided to some degree but not fully to enable for example to reproduce the figures or for example re-calculate the antibody tires (the method how they were calculated is also missing) and repeat statistical analyses.

Conclusions are clearly stated.

Reviewer 3 ·

Basic reporting

The manuscript is well written from the abstract to the discussion. The authors have provided the background of the previous studies, the aim of this study and they have cited the previous research properly.

Experimental design

This study is meaningful and has impact on the vaccine design against Buruli ulcer. The authors have designed and described the methods in a systematic way which is enough for the replication of this work.

Validity of the findings

The authors have validated their work in a systematic way and they have mentioned the limitation of their study and they have cited the literature in a appropriate place. the finding has been validated by the statistical methods and they have considered different statistical significance level for the validation their study.

The conclusion is well proved by the experimental setup and have great significance for the vaccine design.

Additional comments

Manuscript is well designed and written in a systematic way . So, I will recommend to accept the manuscript.

---

## Round 0.2 · accepted · Accept

Your revised manuscript clearly addressed the comments raised by reviewers. I will be looking forward to receiving new works from you in the near future.